# Accuracy Assessment of the Planar Morphology of Valley Bank Gullies Extracted with High Resolution Remote Sensing Imagery on the Loess Plateau, China

**DOI:** 10.3390/ijerph16030369

**Published:** 2019-01-28

**Authors:** Yixian Chen, Juying Jiao, Yanhong Wei, Hengkang Zhao, Weijie Yu, Binting Cao, Haiyan Xu, Fangchen Yan, Duoyang Wu, Hang Li

**Affiliations:** 1State Key Laboratory of Soil Erosion and Dryland Farming on the Loess Plateau, Institute of Soil and Water Conservation, Chinese Academy of Sciences and Ministry of Water Resources, Yangling 712100, China; chenyixian14@mails.ucas.edu.cn (Y.C.); yhweigo@163.com (Y.W.); 2University of Chinese Academy of Sciences, Beijing 100049, China; 3Institute of Soil and Water Conservation, Northwest A&F University, Yangling 712100, China; zhkang1024@hotmail.com (H.Z.); yuweij@nwsuaf.edu.cn (W.Y.); anhuibinting@gmail.com (B.C.); 18704895947m0@sina.cn (H.X.); yanfc1991@163.com (F.Y.); 15389081751@163.com (D.W.); 18821637167@163.com (H.L.)

**Keywords:** valley bank gully, Pleiades imagery, visual interpretation, accuracy assessing standard, gully detection limit, gully dimensions, vegetation cover, gully shape complexity, broken topography

## Abstract

Gully erosion is a serious environmental problem worldwide, causing soil loss, land degradation, silting up of reservoirs and even catastrophic flooding. Mapping gully features from remote sensing imagery is crucial for assisting in understanding gully erosion mechanisms, predicting its development processes and assessing its environmental and socio-economic effects over large areas, especially under the increasing global climate extremes and intensive human activities. However, the potential of using increasingly available high-resolution remote sensing imagery to detect and delineate gullies has been less evaluated. Hence, 130 gullies occurred along a transect were selected from a typical watershed in the hilly and gully region of the Chinese Loess Plateau, and visually interpreted from a Pleiades-1B satellite image (panchromatic-sharpened image at 0.5 m resolution fused with 2.0 m multi-spectral bands). The interpreted gullies were compared with their measured data obtained in the field using a differential global positioning system (GPS). Results showed that gullies could generally be accurately interpreted from the image, with an average relative error of gully area and gully perimeter being 11.1% and 8.9%, respectively, and 74.2% and 82.3% of the relative errors for gully area and gully perimeter were within 15%. But involving field measurements of gullies in present imagery-based gully studies is still recommended. To judge whether gullies were mapped accurately further, a standard adopting one-pixel tolerance along the mapped gully edges was proposed and proved to be practical. Correlation analysis indicated that larger gullies could be interpreted more accurately but increasing gully shape complexity would decrease interpreting accuracy. Overall lower vegetation coverage in winter due to the withering and falling of vegetation rarely affected gully interpreting. Furthermore, gully detectability on remote sensing imagery in this region was lower than the other places of the world, due to the overall broken topography in the Loess Plateau, thus images with higher resolution than normally perceived are needed when mapping erosion features here. Taking these influencing factors (gully dimension and shape complexity, vegetation coverage, topography) into account will be favorable to select appropriate imagery and gullies (as study objects) in future imagery-based gully studies. Finally, two linear regression models were built to correct gully area (*A_ip_*, m^2^) and gully perimeter (*P_ip_*, m) visually extracted, by connecting them with the measured area (*A_ms_*, m^2^) and perimeter (*P_ms_*, m). The correction models were $A_{ms}=1.021A_{ip}+0.139$ and $P_{ms}=0.949P_{ip}+\;0.722$, respectively. These models could be helpful for improving the accuracy of interpreting results, and further accurately estimating gully development and developing more effective automated gully extraction methods on the Loess Plateau of China.

## 1. Introduction

Gully erosion is one of the main manifestations of land degradation [1]. It is defined as the process whereby runoff water accumulates and often recurs in narrow channels and, over short periods, removes the soil from this narrow area to considerable depths [2]. Further, gully erosion is considered as the most destructive form of erosion [3], which causes landform dissection and consequently damages and reduces agricultural land [4], results in land degradation by reducing soil fertility [4] and accelerates aridification [5]. It also affects flow regimes, sediment budgets and deposition rates [6], and induces the silting up of downstream riverbeds and reservoirs [7,8] and thus may cause catastrophic flooding episodes and pollution [9,10]. Moreover, gully erosion undermines infrastructures and alters transportation corridors as well [4,11]. Data collected in different parts of the world show that soil loss by gully erosion represents 10% to 94% of total sediment yield caused by water erosion [2], and contributes 60% to 90% of total sediment production on agricultural land in the hilly and gully areas of the Chinese Loess Plateau [12]. Mapping gullies is thus crucial, for assisting in understanding the magnitude and mechanism of gully erosion, monitoring and predicting its development processes, and assessing its current and future impacts.

Numerous studies have adopted different techniques and methods to map and monitor gully erosion features. In the beginning, conventional field methods mainly including ruler, pole, tape, microtopographic profilers and pins [13,14,15,16] were used, whereas they are time consuming and labor intensive for achieving high accuracy in field surveys and consequently are always confined at smaller spatial scales [1]. Afterwards, with the development of technology, total stations [16,17], differential global positioning system (GPS) [18,19], airborne and ground-based Light Detection and Ranging (LiDAR) systems [11,20], three-dimensional laser scanners [21,22] and remote sensing imagery [1,3,21,23] have been applied to map gullies. These emerging methods obviously improve work efficiency and guarantee the measurement accuracy simultaneously. However, except for that based on remote sensing imagery, they still to some extent face the challenges in time, labor and study spatial scales. And, airborne and ground-based LiDAR systems and three-dimensional laser scanners are also sensitive to vegetation [20] and weakened by limited scanning coverage and side-looking orientation especially when dealing with deeply incised gullies [11]. In recent years, unmanned aerial vehicles (UAV) [24,25,26] have been widely explored for their feasibility in monitoring and quantifying gully erosion, but they also exist similar shortages to the above, despite their advantage of acquiring imagery with a very high level of details [24,25,26]. By contrast, remote sensing imagery has been proved to be a more practical tool, permitting the coverage of large areas with a minimum time and effort [27] and providing satisfactory surface details with the increased availability of high-resolution satellites such as Ikonos (with 1.00 and 4.00 m spatial resolution in panchromatic and multi-spectral mode, respectively), QuickBird (with 0.61 and 2.44 m spatial resolution in panchromatic and multi-spectral mode, respectively) and Pleiades [26,28]. Therefore, so far, remote sensing-based mapping has become the most practical approach for delineating gully features over large areas, given the variability in gully size, shape and occurrence [29]. Meanwhile, remote sensing-based gully mapping is also supposed to be promising in future, with the further improvement to gain imagery with higher spatial-temporal resolution and more other useful information like spectral, stereopair.

Mapping and monitoring gully erosion features from remote sensing imagery have been performed mainly through techniques ranging from visual aerial photograph or satellite image interpretation [1,21,30] to pixel-based image classification analysis (PBICA) [23,31,32] and to object-oriented analysis (OOA) [29,33,34,35]. Among them, pixel-based image classification methods usually base only on the surface reflectance values, causing their performances to strongly depend on the selection of training pixels and the separability of spectral signatures [33,36]. However, spectral heterogeneity exists not only among the gullies themselves but also among their surroundings [37], and which varies easily with the variability in moisture, organic matter and mineral content, and also shadow and atmospheric activities. Hence, the methods changing analysis unit from pixels to objects are superior to the traditional pixel-based methods and also necessary. Object-oriented methods segment pixels into relatively homogeneous regions, i.e., gullies in this case, by integrating spectral, topographic, shape and textural information based on the homogeneous criteria [38,39], which fundamentally resemble the cognitive-based visual interpretation, but in a controlled and reproducible quantitative manner [33,38,39]. Nevertheless, multi-type auxiliary information and complex workflows are essential for the object-recognition. On the one hand, topographical variables (such as slope, specific catchment area, flow direction and roughness; [33,39]), vegetation condition (such as Normalized Difference Vegetation Index—NDVI; [33,34]) and geometric properties of gullies (such as shape, dimension, orientation; [33]) are used as input datasets and segmentation criteria to extract gullies. On the other hand, the main procedures building the workflow, contain data preparation, segmentation, metrics calculation and gully mapping and validation, and so on, in which even more sub-steps are involved [29,33,34]. In addition, these procedures may either strongly interlink with each other or be clearly independent [38]. Accordingly, the object-oriented methods tend to be restricted to the areas with sufficient basic data and knowledge about the gully characteristics, and their accuracy may also be susceptible due to the manifold working procedures and influencing factors that possibly occur in any of the procedures. In general, although the pixel-based and object-oriented methods can semi- or all-automatically detect and delineate gullies from multispectral imagery, the quality of them is generally moderate [30] (e.g., the underestimation of gully area acquired based on PBICA in the Brazilian Cerrados was 25.2% [23], the average area misestimation of the gully system for three sub-watersheds in Morocco based on OOA was 16.2 ± 23.1% [35] and error assessment of gully extent extracted by OOA in Australia’s Tropical Rivers presented an overall accuracy of approximately 50% [29]), and their applicability is limited to the area used for calibration and validation [30]. Moreover, some DEM-based topographic methods, considering the relationship between topographic attributes and gully distribution, have also been designed for gully feature extraction [40,41]. However, they are also accuracy-limited due to be sensitive to terrain characteristics of study area (e.g., the misestimation of gully area in four study areas in Spain ranged from 13.6% to 24% [40] and gully width was overestimated by up to 30% in the Bleaklow Plateau of the UK [41]), consequently which need to be improved further when applied in different regions [42]. As a result, gully detection and delineation on images is still done visually and manually in most studies because of the convincing accuracy of this way. Visual interpretation of remote sensing imagery is an old and common method for feature extraction [43]. Commonly, the development of automated feature extraction schemes is preceded by visual examination of scene characteristics [43], which provides training and validation datasets for them [23,29,30,33,35,39].

Although the results extracted by visual interpretation have been widely used in the development of newly automated methods, the accuracy of them seems to be rarely assessed first (see, e.g., [23,33,34]), as the same case in gully evolution studies where the gullies of different time periods visually interpreted from images were directly compared (see, e.g., [44,45]). It appears that the impressions, that the gullies delineated visually and manually from images could straight be evaluated without judging the accuracy themselves first and that the images with higher resolution would extract gullies more accurately, have been gradually created by the present studies. It is undeniable that remote sensing images having finer resolution generally generate more precise consequences [30,35], but to what extent could gullies be accurately delineated and what is the minimum dimensions of gullies that could be accurately delineated both from remote sensing imagery with high resolution? And how do the potential factors affect delineating accuracy and is that possible to further improve the accuracy of delineated gullies by connecting them with the actual gullies? Attempting to figure out these questions is helpful, for better utilizing the high-resolution satellite images with increasing availability; and for contributing to the gully erosion studies at larger spatial scales, especially under the imperative that the soil erosion and land degradation affected by the global climate change and human activities at regional and global scales should be assessed, by providing more accurate validation datasets to develop more effective automated gully extraction methods.

In this study, an image from the Pleiades that is a follow-up satellite of the SPOT (Système Pour l’Observation de la Terre) was selected and assessed, because it is a good consideration with both very high spatial resolution and acceptable price for most of the scientific community. The objectives of this study are to (1) evaluate the potential of Pleiades image to delineate gully erosion features through visual interpretation, (2) explore the influence of potential factors on visually gully interpreting accuracy and (3) establish the correction models between visually interpreted and field-measured gully morphology parameters for a given typical area in the Chinese Loess Plateau.

## 2. Materials and Methods

### 2.1. Study Area and Gully Types

The study area, the Fangta watershed (109°14′42′′–109°16′59′′ E, 36°46′21′′–36°49′30′′ N), is located in Ansai County, northern Shaanxi Province, and belongs to the hilly and gully region of the Loess Plateau (Figure 1a,b). The watershed covers an area of 10.54 km^2^ with elevations ranging from 1048 to 1363 m with an average of 1206.53 ± 58.55 m (mean ± standard deviation, hereinafter the same). Average slope gradient is 26.44 ± 13.21°, and the area of slopes with slope gradient >25° comprises 53.34% of the entire watershed. This watershed is characterized by a warm, temperate-continental monsoon climate with an annual average temperature of about 8.80 °C [46]. The average annual rainfall is 528.48 mm from 1971 to 2014, more than 70% of the annual total rainfalls fall between June and September and are often associated with storms, which are typically characterized by short duration and high intensity [47].

The study area belongs to the forest steppe ecotone [48]. Due to the implementation of the Returning Farmland to Forest and Grassland Program since 1999 with the major measures of reforestation and abandonment of farmlands with slope above 25°, vegetation has been efficiently recovered with the total area of woodlands and grasslands beyond 70% of the watershed area in 2014. The dominant species include *Robinia pseudoacacia*, *Caragana intermedia*, *Hippophae rhamnoides*, *Sophora viciifolia*, *Artemisia gmelinii*, *Artemisia giraldii*, *Lespedeza davurica* and *Bothriochloa ischaemun*. Loessial soil is the main soil type in the watershed and is mainly composed of 65% sand, 24% silt and 11% clay by weight [49]. Because of the loose soil particles and the poor erosion resistance, the soil is prone to erosion [8].

The study area shows a landscape of high-density and deep-cut gullies with a gully density of 3.8 km/km^2^ [50], and consequently an extremely broken and undulating topography. Three main gully types exist on the Chinese Loess Plateau: hillslope gully, valley floor gully and valley bank gully [19] (Figure 2). Hillslope gullies mainly develop in the cropland portion of the interfluves. They generally have cross-sections of ‘U’ shape in the early formation stage and which gradually evolve as ‘V’ shape. Valley floor gullies develop on the bottom of the valley. Their cross-sections also present ‘U’ shape at first but grow into trapezoid shape later [51]. This two gully forms are both characterized by a long-narrow planar morphology (with width usually less than ten meters) [51]. Valley bank gullies occur continuously at the boundaries between interfluves and valleys [19] (Figure 1c, Figure 2a). They exhibit a short-deep morphology, which is significantly deeper than the above two gully types. Furthermore, valley bank gullies are the most important sources of sediment yield to river and the main threat to inter-valley land quantity and quality [21]. Therefore, this study focused on valley bank gullies, which were selected based on the following steps. First, on the basis of the satellite image (more details see Section 2.2), a transect was placed randomly in and throughout the study watershed, and seven valleys were selected along the transect (Figure 1b). Then, in the field, a total of 130 valley bank gullies, which occurred on the boundaries of the seven valleys, were selected as the study objects according to their dimensions and morphology that were representative in the study watershed.

### 2.2. Pleiades Image

Pleiades-1B image contains one panchromatic band with 0.5 m resolution and four multi-spectral bands with 2.0 m resolution. Its extensive image swath width (20 km) allows gully mapping over large regions. The image was specially acquired in the early winter (21 November 2013), because the vegetation cover was lower at that time compared with the other seasons of the year, resulting from the withering and falling of vegetation, and without snow cover yet as well, under which the edges of gullies could be distinguished and traced well [52]. After the geometric, radiometric and atmospheric corrections strictly performed by the data provider (Airbus Defence and Space, Toulouse, France), the panchromatic and multi-spectral images were fused to generate a 0.5 m pan-sharpened image using the software ENVI 5.3 (Exelis Visual Information Solutions, Inc., Broomfield, CO, USA). Subsequently, the gully edge lines (Figure 1c) would be extracted by visual interpretation from the processed image.

### 2.3. Field-Measured Data

Field-measured gully edge lines were obtained using a SOUTH S86 RTK (South Surveying & Mapping Technology CO., LTD, Guangzhou, China) differential GPS system: with planimetric and altimetric precision of 1 cm ± 1 ppm and 2 cm ± 1 ppm, respectively. Turning points of gully edges were successively measured and numbered using the GPS in the field, which was conducted in July 2014. The measurement intervals between two contiguous points were mainly within 0.2–1.4 m (comprising 80% of all the intervals), for some parts of gully edge with intricate morphology, more intensive measurements were taken. After the field measurements, the collected point data were transformed into gully edge lines by connecting the points in order with line in the software ArcMap 10.3 (Esri, Redlands, CA, USA).

It is noteworthy that the period between the image and field data acquisition periods (December 2013–June 2014) is regarded as insignificant for the purposes of this study, because gully erosion mainly occurs during the rainy season, i.e., June to September, due to the driving force by rainfall to trigger and accelerate soil erosion [4,53]. Moreover, the land uses in upslope drainage area of the study valleys were woodlands or grasslands (except that one of them was terrace abandoned before November 2013) in 2013 and did not change so far, and grazing was banned in the area from March 2008, which both greatly reduced the possibility of gully formation and expansion caused by human disturbances and grazing during the above period.

### 2.4. Gully Interpreting from Image

Visually interpreting the gullies from the processed image was carried out in ArcMap by a trained PhD-level operator based on the spectral signatures and the spatial characteristics of valley bank gully, i.e., shape, size, patterns and contextual information. The operator has been being involved in projects dealing with gully erosion in the Loess Plateau region for more than 4 years and has participated in field investigation and indoor interpretation of gullies many times, having an in-depth knowledge of geomorphology and gully erosion in the study area, and possessing the necessary background and ability to conduct the mapping. In addition, because of the gullies studied here are valley bank gully, which specially occur at the boundaries between interfluve and valley and have relatively homogeneous dimensions and forms, not like those in the similar studies having significant variation in scale [26,52] and being easily confused with other surface features such as furrows, wheel tracks and irrigation channels, because those studies usually focused on gullies occurring on agricultural land zones [3,30]. Therefore, the interpretation signs and decision tree like such in previous studies [26,30,52], did not need to be established first in this study, but the gullies be interpreted directly relying on the expertise of the operator. In this study, the gullies were repeatedly interpreted three times by the operator, and in each time every gully was mapped, in this way, the mapped replication 1, 2, 3 of each gully were obtained. Doing this was to improve the interpreting accuracy by using the average of the three replications as the interpreted parameters eventually compared with measured values, and to evaluate the systematic deviation of mapping gullies by the operator. If the systematic errors produced by the same operator had been generally larger than the interpreting errors (absolute errors) between the mapped and the measured gullies, this study would have been valueless.

After that, the morphological parameters of interpreted and measured gullies, including gully length, gully width, gully area and gully perimeter, were calculated based on the gully edge lines using ArcMap and AutoCAD F.51.0 (Autodesk, Inc., San Rafael, CA, USA).

### 2.5. Assessment of Interpretation Accuracy

In this study, gully area and gully perimeter were selected as the gully morphology parameters analyzed in accuracy assessment, because only this kind of planar morphology parameters could be extracted from the imagery without stereopair, but also because gully area and gully perimeter are common and important parameters assessed in the gully erosion studies [16,54,55]. On the one hand, gully area and gully perimeter over different periods are direct indicators to understand gully development processes and to quantitatively assess gully erosion. Especially when at large spatial scales, the gully depth information is scarce and costly to gain, but the gully erosion studies at these scales are imperative under the intensive global climate change and human activities. On the other hand, gully area has been proved to be an effective geometric variable to predict gully volume in the Loess Plateau [21,56], thus estimating erosion volumes and conducting some more in-depth gully erosion studies were also possible based on this parameter. Moreover, before assessing the interpreted gullies with measured gullies, systematic deviation of delineating the gullies by the operator was evaluated first. Hence, the systematic error of gully area (*SE_A_*) and gully perimeter (*SE_P_*), i.e., the average of the differences of each two interpreted gully replications, and the absolute error of area (*AE_A_*) and perimeter (*AE_P_*) between the interpreted and the measured gully, were quantified as follows:(1)$$A_{ip}=\frac{1}{3}\left(A_{ip1}+A_{ip2}+A_{ip3}\right)$$
(2)$$P_{ip}=\frac{1}{3}\left(P_{ip1}+P_{ip2}+P_{ip3}\right)$$
(3)$$SE_{A}=\frac{1}{3}\left(\left|A_{ip1}-A_{ip}\right|+\left|A_{ip2}-A_{ip}\right|+\left|A_{ip3}-A_{ip}\right|\right)$$
(4)$$SE_{P}=\frac{1}{3}\left(\left|P_{ip1}-P_{ip}\right|+\left|P_{ip2}-P_{ip}\right|+\left|P_{ip3}-P_{ip}\right|\right)$$
(5)$$AE_{A}=\left|A_{ip}-A_{ms}\right|$$
(6)$$AE_{P}=\left|P_{ip}-P_{ms}\right|$$
where *A_ip_*_1_, *A_ip_*_2_ and *A_ip_*_3_ are area of the interpreted gully replication 1, 2 and 3, respectively; *P_ip_*_1_, *P_ip_*_2_ and *P_ip_*_3_ are perimeter of the interpreted gully replication 1, 2 and 3, respectively; *A_ip_* and *P_ip_* are the average of the interpreted gully area and gully perimeter, respectively; *A_ms_* and *P_ms_* are the measured gully area and gully perimeter, respectively.

After that, the relative error of area (*RE_A_*) and perimeter (*RE_P_*) between the interpreted and the measured gully were quantified as follows:(7)$$RE_{A}=\frac{\left|A_{ip}-A_{ms}\right|}{A_{ms}}\times 100\%$$
(8)$$RE_{P}=\frac{\left|P_{ip}-P_{ms}\right|}{P_{ms}}\times 100\%$$

### 2.6. Factors Potentially Influencing Gully Interpreting

To understand the influence of different factors on gully interpreting accuracy, potential factors including gully dimensions, gully sinuosity and the vegetation cover surrounding gully edge were gained with the following methods. The dimensions of gully consisted of length (*L_ms_*), width (*W_ms_*), area (*A_ms_*) and perimeter (*P_ms_*), which were all field measure-based and had been calculated in Section 2.4. The vegetation coverage (represented with NDVI) of the study watershed was first derived from the pre-processed Pleiades image with the methods same as Rouse, et al. [57] and Qi, et al. [58]. Then, outside 5 m buffer zones along gully edge line were created in ArcMap, which were further used to extract vegetation cover surrounding gully edge (*VC*) from the watershed NDVI image. Gully sinuosity (*Sin*) was defined as the ratio between the perimeter of a gully and the perimeter of a circle of equal area [59], which reflected the shape complexity of gully edges:(9)$$Sin=P_{ms}/P_{Circ}$$
(10)$$P_{Circ}=2\pi \cdot {\left(A_{ms}/\pi \right)}^{1/2}$$
where *P_ms_* and *A_ms_* are the measured gully perimeter and gully area, respectively; *P_Circ_* is the perimeter of a circle having equal area with the gully.

### 2.7. Software and Methods for Data Analysis

Influence of different factors on gully interpreting accuracy was analyzed by Spearman correlation analysis using R 3.5.0 (R Development Core Team, Vienna, Austria). Linear regression was used to establish the correction models between measured and interpreted gully area and gully perimeter using SPSS 23.0 (International Business Machines Corporation, Armonk, NY, USA). The figures were plotted with Origin 2018C (OriginLab Corporation, Northampton, MA, USA) and “ggplot2” package [60] written for R 3.1.0.

## 3. Results

### 3.1. Accuracy Assessment of Interpreted Gullies

The results of the Paired-Samples *T* Test between *SE_A_* and *AE_A_* and between *SE_p_* and *AE_p_* demonstrated that the absolute errors of area and perimeter were both significantly higher than their respective systematic errors (*p* < 0.01, Table 1), indicating that the subsequent analyzing and discussing of this study were meaningful.

Therefore, it could be seen from Figure 3 that the relative errors of gully area and gully perimeter varied from 0 to 53.7% and 0.1% to 47.8%, with an average of 11.1% and 8.9%, respectively. The majority of relative errors for area and perimeter were both within 15%, accounting for 74.2% and 82.3% of all the corresponding errors, respectively. Generally, the accuracy of interpreted gullies was maintained at a high level.

### 3.2. Influence of Potential Factors on Gully Interpreting Accuracy

The correlation matrix was used to analyze the influence of potential factors on the interpreting accuracy of gully (Figure 4).

The results showed that the parameters representing gully dimensions were all significant negatively correlated with the gully interpreting errors (*p* < 0.01), except that the correlation between *W_ms_* and the perimeter-related errors were just significant at the 0.05 level, which indicated that larger gullies could be interpreted more accurately from remote sensing imagery and vice versa. The *Sin* was significant positively correlated with all the errors (*p* < 0.01), resulting from larger possibilities to lose details of gully edge during interpreting the gullies with higher *Sin*. However, *VC* had no significant correlation with all the errors (*p* > 0.05), which might be attributed to the generally lower vegetation coverage (0.19 ± 0.11) when the interpreted image was acquired.

### 3.3. Correction Model of Interpreted Gully Morphology Parameters

Linear regression models between interpreted and measured gully area and gully perimeter were respectively fitted as follows:(11)$$A_{ms}=1.021A_{ip}+0.139\;(n\;=\;130,\;R^{2}\;=\;0.994)$$
(12)$$P_{ms}=0.949P_{ip}+0.722\;(n\;=\;130,\;R^{2}\;=\;0.983)$$
where *A_ms_* is measured gully area (m^2^) and *A_ip_* is interpreted gully area (m^2^), *P_ms_* is measured gully perimeter (m) and *P_ip_* is interpreted gully perimeter (m).

Actually, in order to take the regional variability in vegetation cover and gully shape complexity into account, *VC* and *Sin* were also used as independent variables together with *A_ip_* and *P_ip_* to try building relationship with *A_ms_* and *P_ms_*, respectively, but they both did not entered into the resulting equations in the stepwise multiple regression, so *A_ip_* and *P_ip_* were then directly connected with *A_ms_* and *P_ms_*, respectively. The two fitted equations both had considerably large values of coefficient of determination (*R*^2^), with 0.994 for that of gully area and 0.983 for that of gully perimeter, which indicated that *A_ip_* and *P_ip_* were enough to predict the actual gully area and gully perimeter accurately, respectively. To validate the models further, the predicted gully area and gully perimeter values were respectively compared with their measured values (Figure 5). It could be observed that the measured values and the predicted values were distributed along the 1:1 line, and the trend lines for area and perimeter both almost entirely overlapped with corresponding 1:1 line, with a *R*^2^ value of 1.000 for area trend line and 0.983 for perimeter trend line. These demonstrated that the two correction models were reliable. Although it could be found that the interpreted gully area seemed not to be transformed so much before into coincident values of measured gully area through Equation (11), considering the fact that the average relative error of interpreted gully area was 11.1% and 25.8% of all the study gullies had *RE_A_* greater than 15%. Thus, the established relationship between the interpreted and measured gully area still presented practical value for correcting gully area extracted based on satellite images in this area.

## 4. Discussion

### 4.1. A quantitative Standard to Judge Gully Delineating Accuracy

Previous studies extracting gully erosion features based on remote sensing images rarely determined an exact standard to judge whether a gully was delineated accurately, however, which was important for assessing the ability of remote sensing images to interpret gully and further for conducting gully evolution research at multiple spatial-temporal scales. Here, inspired by Evans and Lindsay [41] who found errors of delineated gully width always close to the resolution of the used imagery, a potential standard adopting one-pixel tolerance along gully edge line to assess gully interpreting accuracy from remote sensing images was proposed in this study.

Due to gully edges being defined as the break of slope and usually at which the bare soil and vegetation cover were also divided. In addition, pixel is the smallest unit of an image (for this study its dimension is 0.5 m × 0.5 m), and it is fully uniform in interior even if there should have been some smaller ground details existed. Thus, the outward sides from gully inside of these pixels which spanned the gully edge would be assigned to the gully during image interpretation. However, in fact, gully edges were not necessarily the outward sides of the break pixels, it could be anywhere in each break pixel if the gully edge was generalized as a line with width sufficiently small when compared with that of a pixel and, considering that a pixel is a uniform whole in which the specific position where gully edge line occurred was unknown. So, this is the reason why the tolerance was specified as one image pixel. The area tolerance (*AT*) for each gully in this study was obtained by creating 0.50 m buffer polygon along the interpreted gully edge line and then calculating the area of them. Further, the area tolerance was compared with corresponding absolute area error (*AE_A_*) gully by gully. Finally, the gullies, having the ratio of *AE_A_* to *AT* (*AE_A_*/*AT*) within 1.00, were supposed to be delineated accurately because their interpreting errors did not exceed the corresponding tolerances, and the closer the value of *AE_A_*/*AT* to zero, more accurately the gullies were delineated. The division of gully interpreting accuracy using the *AE_A_*/*AT* index was presented in Table 2. The results showed that more than 80% of the gullies were accurately mapped in this study, indicating that high-resolution remote sensing image had the potential to extract gully overall precisely by the means of visual interpretation. This is also in accordance with the results in Section 3.1.

Hence, it could be concluded that the proposed standard adopting one-pixel tolerance along the interpreted gully edges was able to effectively judge gully interpretation accuracy. Besides, this standard could be readily applied to the studies with regard to gully mapping based on currently popular UAV imagery and aerial photograph as well (provided that the data were preprocessed strictly) whenever manual digitizing is involved, since this two types of images had similar acquisition mechanisms and nature with satellite image. Nevertheless, this standard should be carefully used for those smaller gullies or coarser resolution images, because that would lead to relatively greater proportion of one-pixel tolerance area compared to the gully area [41].

### 4.2. Gully Width Detection Limit of Remote Sensing Imagery in the Loess Plateau

The spatial resolution of imagery is an indication of the potential details of the features in the imagery and consequently strongly influences the level of gully detectability [3,35,41,56,61]. Generally, it was possible to identify smaller gullies in the finer resolution images than in the coarser resolution data [28,35,41], as was shown in Figure 6a where a positively linear relationship between image resolution and minimal identifiable gully width was exhibited. Moreover, utilizing the images with a better resolution would likely decrease the uncertainty in gully detection and delineation [30], accordingly, the inaccuracy of the gully features extracted increased with a decrease in image resolution [35].

In Figure 6a, it could be found that all the points distributed above or at the 1:1 line at least, indicating that remote sensing imagery could identify gullies whose width was equal to or greater than the imagery resolution, which was consistent with those concluded by Desprats, et al. [3], Vrieling, et al. [23], Vrieling, et al. [65], and Warren, et al. [66]. In this study, the lower limit of width of accurately delineated gullies (3.18 m), which were determined according to the standard proposed in Section 4.1, significantly larger than those detected from images with same resolution (0.50 m) in previous studies (Figure 6). This may be caused, on the one hand, by the inherent magnitude of permanent gullies in the study area (with a range of gully width from 3.18 to 35.23 m and an average of 10.05 m for this study), which was a result of the combination of frequent rainstorms and typical soil properties prone to erosion in this region [15]. However, it could also be observed that from Figure 6b, in this study, the width range of inaccurately delineated gullies was within that of accurately delineated, indicating that the image used here should have been able to discern the incorrectly delineated gullies.

This was possibly related to the broken topography in the study area (Figure 7), which was inferred through reviewing the overlay of the incorrectly discerned gullies on the true-color Pleiades image. Some gullies could not be accurately delineated even with a larger scale due to the broken topography, this was, on the other hand, the reason why the minimal identifiable gully width in this study larger than those in literature. Consequently, the gully detection limits based on satellite image in this region was narrower compared to those in other regions around the world, images with higher resolution than normally perceived thus were needed when mapping erosion features here, especially for those with smaller dimensions.

### 4.3. The Causes of Higher Gully Interpreting Errors

Through overlaying the mapped gully edge lines with *RE_A_* and *RE_p_* respectively >15% on the true-color Pleiades image, it was found that the higher interpreting errors were mainly caused by broken topography, vegetation cover and the deviation in measured data, as presented in Table 3 and Figure 7 and Figure 8. Firstly, the broken topography, on the one hand, was a result of the long-term and serious gully erosion in this area, especially that of permanent gully [15]. On the other hand, the three cases in Figure 7, all made great contributions to the formation of such topography, meanwhile, they left multiple and complex slope break lines on loess surfaces, which seemed extremely like gully edges on the satellite image and consequently confused the interpreters with the actual gully edges during delineating [67]. Secondly, although vegetation cover was not significantly correlated with errors of all types in this study due to its overall lower values in winter, several comparatively high coverage among them still brought about large interpreting errors to their corresponding gullies (Figure 8a). Vrieling, et al. [23] found that a gully had low activity when abundant trees grew in and around it, implying that further erosion of the gully was minimal, and thus the distinct characteristics of exposed bare subsoil was not dominant and moreover concealed by the tree crowns, resulting in the limited success in delineating these gully edges from a QuickBird image in Brazilian Cerrados. Moreover, Marzolff and Poesen [68], James, et al. [20] and Desprats, et al. [3] reported similar findings respectively from aerial photographs taken by a blimp-based remoting sensing platform in semi-arid southeast Spain, LiDAR topographic data in the southeastern United States and QuickBird and SPOT images in Tunisia. Therefore, more emphasis should also be laid on the development of new methods to map the gullies under vegetation from imagery in the future. Then, the deviation of measured gully edge lines mainly occurred in which gully wall was severely undercut near the bottom and thus forming overhanging gully scarp (Figure 8b) that was weak to support weight and accordingly hard to get to as close as possible during the field measuring. This is also the reason why gully mapping based on remotely sensed data are increasingly explored, owing to its non-contact feature [69], of course, also its higher efficiency compared to field measurements [1,17,26,30]. In this study, the available gully area, consuming unitary time, labor and money, for field-based and image-based methods were 10 m^2^/(min·person·dollar) and 167 m^2^/(min·person·dollar), respectively (Appendix A). The latter was about 17 times larger than the former. 

The effect of gully development and shadow just held a small part in the interpreting errors of gullies (Table 3). Among them, gully growth means some gullies had grown when the measured data was obtained compared to that time the image was acquired, which might due to the accidental collapse of gully sidewalls. Shadow hinders the visual interpretation of gully by making the surface invisible or incompletely visible. But Giménez, et al. [27] found that areal gully determination based on photogrammetry was almost unaffected by shadow since under the situation where gully depth was not involved, gully edges could be defined with a much larger accuracy compared to its inner contours.

Figuring out the causes of the higher interpreting errors was not only helpful for understanding the roles that different factors played in gully interpreting based on remote sensing imagery, but also for selecting appropriate imagery indoors and gullies (as study objects) outdoors in the future imagery-based gully studies by taking these factors into account.

## 5. Conclusions

In this study, it was found that gullies could be overall accurately interpreted using the Pleiades satellite image, with an average relative error of gully area and gully perimeter being 11.1% and 8.9%, respectively, and 74.2% and 82.3% of the relative errors for gully area and gully perimeter were within 15%. Still, field measurements of gullies to be involved is a serious recommendation for present imagery-based gully studies. Moreover, a standard adopting one-pixel tolerance along the mapped gully edge to judge whether gullies were mapped accurately was proposed, by which the reliability of the interpreted gullies was similarly confirmed, indicating that this standard was an effective way to judge gully interpretation from remote sensing imagery. Correlation analysis indicated that larger gullies were able to be interpreted more accurately but the increasing gully shape complexity would decrease interpreting accuracy. Overall lower vegetation coverage in winter basically did not affect gully interpretation because vegetation had withered and fallen. Furthermore, due to the overall broken topography in the Loess Plateau, the gully detectability on imagery in this region was lower compared to those in other regions worldwide, consequently images with higher resolution than normally perceived are needed when mapping gully erosion features here. Taking the roles of these influencing factors into consideration will be beneficial to select appropriate imagery and gullies (as study objects) in future imagery-based gully studies. Finally, two models were built for correcting gully area and gully perimeter visually extracted from remote sensing images, by connecting them with the area and perimeter measured (Equations (11) and (12)), respectively. These models will be helpful for improving the accuracy of interpreting results, and further accurately estimating gully development and developing more effective automated gully extraction methods on the Loess Plateau of China.

## Figures and Tables

**Figure 1 ijerph-16-00369-f001:**
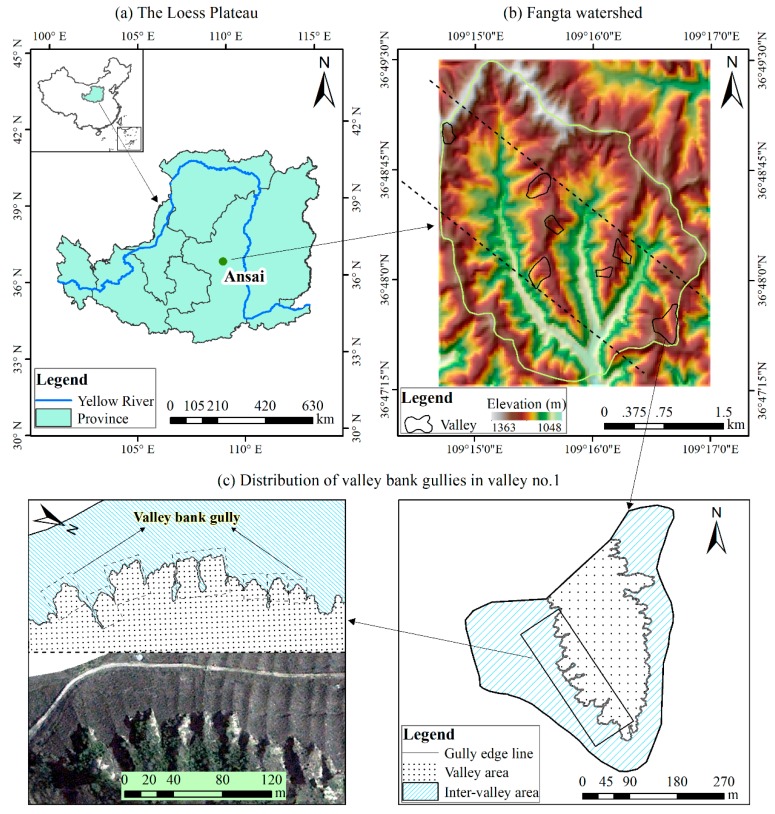
Location of the study area and valley bank gullies there: (**a**) study area on the Loess Plateau in China; (**b**) variation in the elevation of the Fangta watershed; (**c**) visually interpreted valley bank gully edges from Pleiades-1B image in the study area, in which the Pleiades image was displayed as true color composite with a spectral band combination of B2, B1 and B0 assigned to red, green and blue, respectively.

**Figure 2 ijerph-16-00369-f002:**
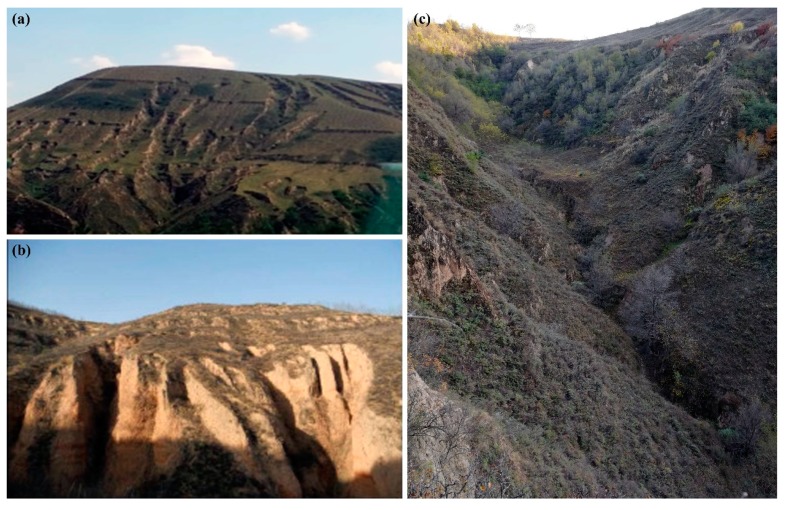
Three main gully types in the Loess Plateau region. (**a**) Hillslope gully, photo by author; (**b**) valley bank gully, photo from Li, et al. [1]; (**c**) valley floor gully, photo by author.

**Figure 3 ijerph-16-00369-f003:**
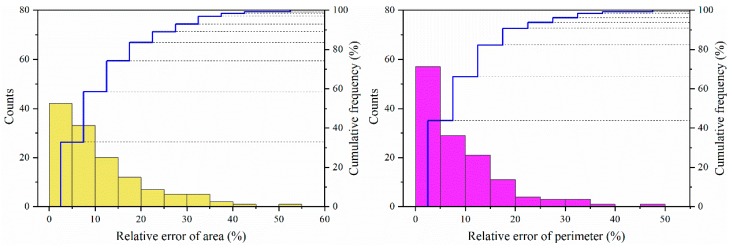
Frequency distribution of the relative errors of gully area and gully perimeter.

**Figure 4 ijerph-16-00369-f004:**
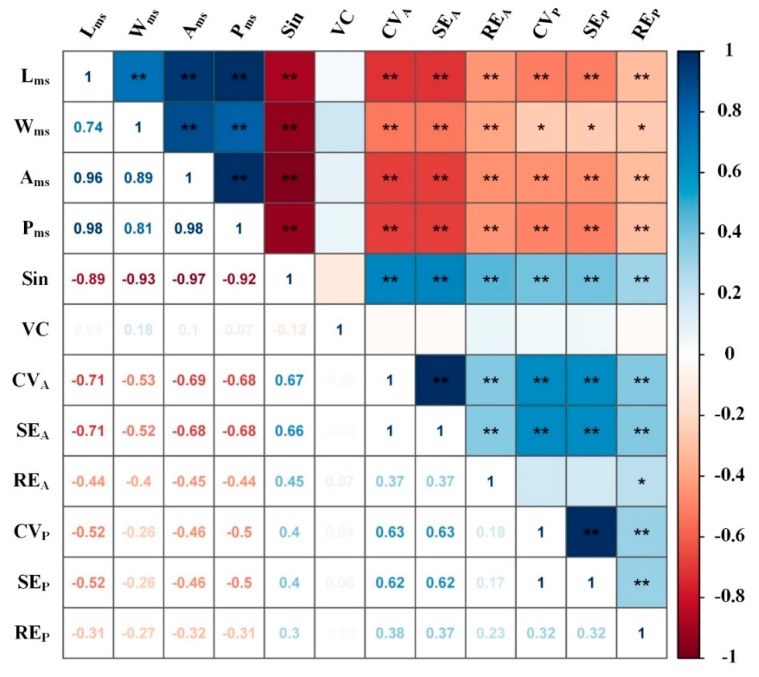
Correlation matrix between gully interpreting errors and potential influencing factors. Note: *L_ms_*, measured gully length; *W_ms_*, measured gully width; *A_ms_*, measured gully area; *P_ms_*, measured gully perimeter; *Sin*, sinuosity of gully edge line; *VC*, vegetation cover; *CV_A_*, coefficient of variance of interpreted gully area replications; *SE_A_*, systematic error of interpreted gully area replications; *RE_A_*, relative error between the interpreted and the measured gully area; *CV_P_*, coefficient of variance of interpreted gully perimeter replications; *SE_P_*, systematic error of interpreted gully perimeter replications; *RE_P_*, relative error between the interpreted and the measured gully perimeter. * Correlation is significant at *p* < 0.05 level (2-tailed). ** Correlation is significant at *p* < 0.01 level (2-tailed).

**Figure 5 ijerph-16-00369-f005:**
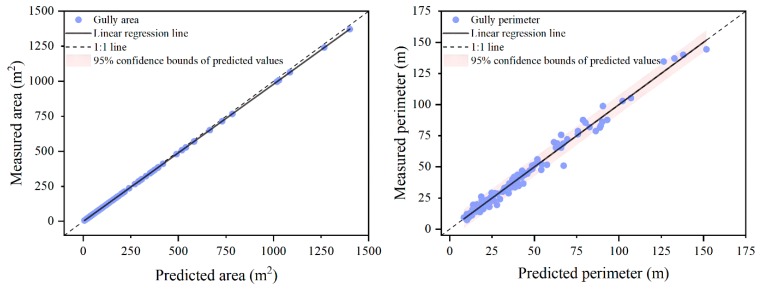
Comparison between the predicted and the measured gully area and between the predicted and the measured gully perimeter.

**Figure 6 ijerph-16-00369-f006:**
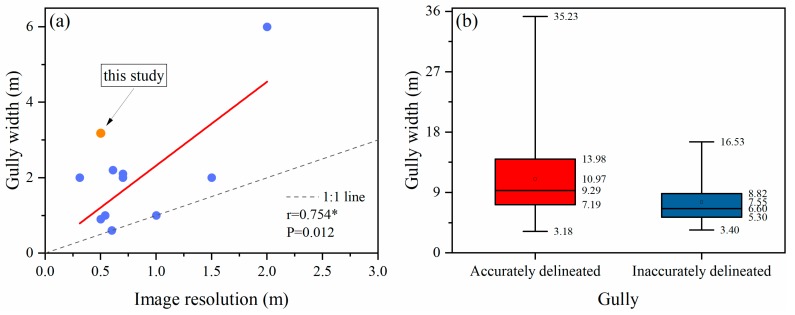
(**a**) The relationship between image resolution and minimal identifiable gully width, which were collected from following literature: [3,11,26,30,35,39,41,62,63,64]. (**b**) The width range of gullies that were accurately and inaccurately delineated from Pleiades image, respectively, which were divided according to the standard determined in Section 4.1.

**Figure 7 ijerph-16-00369-f007:**
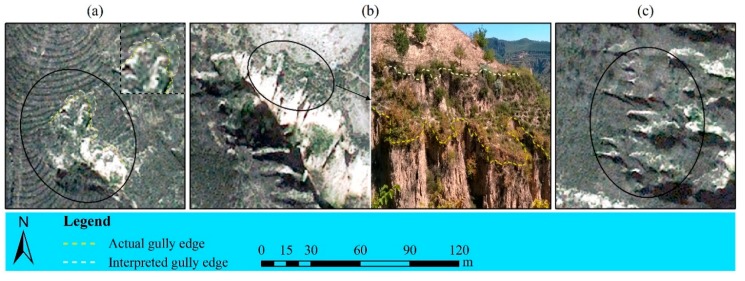
Broken topography in the Loess Plateau mainly caused under three cases. (**a**) Periodical erosion of runoff to original loess surfaces; (**b**) varying anti-erodibility between different loess layers; (**c**) abiogenetic mass failure such as landslide, fault, collapse and so on. The Pleiades image was displayed as true color composite with a spectral band combination of B2, B1 and B0 assigned to red, green and blue, respectively.

**Figure 8 ijerph-16-00369-f008:**
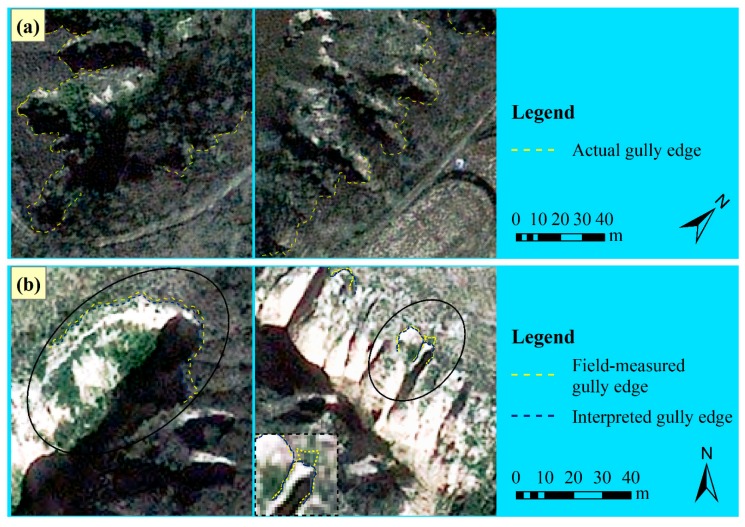
Gullies with dense vegetation cover (**a**) and high and overhanging gully scarps (**b**). Pleiades image was displayed as true color composite with a spectral band combination of B2, B1 and B0 assigned to red, green and blue, respectively.

**Table 1 ijerph-16-00369-t001:** Comparison of systematic error and absolute error of gully area and gully perimeter (mean ± SD).

Area (m^2^)		Perimeter (m)	
Absolute Error	Systematic Error	Absolute Error	Systematic Error
12.8 ± 16.5A	2.8 ± 3.4B	2.9 ± 3.1A	0.9 ± 0.9B

Note: Different letters between different types of errors of area or perimeter indicate significant differences (*p* < 0.01).

**Table 2 ijerph-16-00369-t002:** Division of gully interpreting accuracy based on the *AE_A_*/*AT* index (mean ± SD).

Gully Types	*AE_A_*/*AT*	Percentage (%)
Accurately delineated (*AE_A_*/*AT* ≤ 1.00)	0.42 ± 0.28	80.77
Not accurately delineated (*AE_A_*/*AT* > 1.00)	1.61 ± 0.82	19.23
All	0.65 ± 0.64	100

**Table 3 ijerph-16-00369-t003:** Statistics of main causes to higher gully interpreting errors.

Causes	*RE_A_*		*RE_P_*	
	Percentage (%)	Relative Error (%)	Percentage (%)	Relative Error (%)
Broken topography	26.47	42.20 ± 30.73	36.36	27.46 ± 8.17
Vegetation cover	38.24	25.17 ± 9.18	36.36	24.64 ± 10.18
Deviation in measured data	20.59	24.79 ± 7.58	18.18	20.21 ± 5.49
Gully growth	2.94	20.19	4.55	17.97
Shadow	11.76	18.97 ± 1.69	4.55	18.76

Note: The relative error is reported as the form of mean ± SD (if necessary).

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
