# Peer review of "Accuracy Assessment of the Planar Morphology of Valley Bank Gullies Extracted with High Resolution Remote Sensing Imagery on the Loess Plateau, China"

_ijerph, 2019, doi:10.3390/ijerph16030369_

Round 1

Reviewer 1 Report

The paper concerns the valley bank gully mapping and monitoring with high resolution remote sensing imagery. The collected research material has high quality and work methodology is adequate. The obtained results and proposed empirical models can helpful for accurately estimating gully development and developing more effective automated gully detection methods.

1/ General comments:

I propose to add the table with results of gullies morphology parameters measurement from satellite image and the GPS in the field. It is important for readers because the bank gully forms are different from typical large and complex loess gully form. In my opinion in the paper are described only small erosional forms which represented the first phase of valley bank dissection.

Few specific comments were provided directly on the pdf version (enclosed pdf).

Reviewer 2 Report

Here are few comments on submitted manuscript titled: Accuracy assessment and correction model of gully morphology extracted with high resolution remote sensing imagery on the Loess Plateau, China.

Specific Comments

1.      Line 24: mention the spectral resolution of the image

2.      Line 38-40: mention the algorithm or model used for gully correction.

3.      Line 81: mention the spatial resolution of Ikonons and QuickBird.

4.      Figure 1 (b): Elevation model can be improved visually by changing the colour hue of grey. In legend min and max are not clear.

5.      Line 170: Three gully types are mentioned; hillslope gully, valley bank gully and floor gully. But in rest of the paragraph only one type valley bank gully has been described.  Describe the other two gully types morphologically and their position in catchment area.

6.      Line 201-204: only one gully erosion factor i.e. rain fall has been considered what about land use and human activity in the area which can affect the gully formation and expansion directly impact the accuracy of gully mapping.

7.      Line 208-211: Gully mapping is relied upon the operator’s knowledge and visual interpretation. Gullies have been identified in two dimensions only while visually barren land and gully look like same on image based upon their spectral reflectance due to no vegetation cover however third dimension (i.e. depth information plays a critical role to identify a gully. Please justify without considering the third dimension how gullies can be mapped accurately?

8.      Line 236-238: Define and describe systematic error of area and perimeter as well as absolute error of area and perimeter.

9.      Line 356:  Write UAV stand for what?

General Comments

1.       Background information of the study area needs to be improved.

2.       Presented method is focusing on the active gullies only how this method justifies the accurate mapping of semi stable gullies or gullies with vegetation on edge?

3.          Third dimension needs to be considered for gully mapping that siginificantly impact the accuracy of gully mapping. This point should be justified in this research paper.

Reviewer 3 Report

Thank you for the opportunity to review this interesting paper.  I have made detailed comments on the PDF version of the paper.   

The title is too long.  Better = “Valley Wall Gully morphology on the Loess Plateau, China: Comparing field and remotely sensed measurements”  

Overall this is a useful study but I have three general points:

1.      The study is ONLY concerned with the gully lip or upper edge.  It does not appear to cover gully depth, volume, or cross-sectional form.  In terms of estimating sediment volumes, this is the critical variable.  So this study actually covers a limited aspect of gully morphology.  The title should reflect this, and the paper needs to cover the benefits of just measuring the expansion of the gully edge.  It also only covers valley wall gullies rather than all gully types, which should also be reflected in the title. 

2.      An aspect of this study that is not covered is safety.  I cannot believe that it was safe to measure the edges of those valley wall gullies in the field.  How close were people really willing to get?  I imagine that prudence is a larger source of error than anything else.  However, this is also a strong argument for using remotely sensed measurements.  Otherwise how would one really measure these extreme gullies? 

3.      The quality of the discussion needs to be improved.  What is the generality of the results here?  How exactly is this project an advance on earlier studies that were similar?   Is there something special about the satellite used?  About the methods used?  Exactly which gap in the literature does the paper fill?  

Quality of the English is not too bad.  It does need a detailed review by a native English speaker however. 

Only requires one significant figure for percentage estimates.  Not realistic to have two. 

The references need to be checked throughout as proper nouns (like place names) do not have capital letters at the front.  Also many spelling errors in the bibliography

A useful addition to the paper would be an estimate of the relative cost of field and satellite measurements (expressed as cost per kilometre or area reviewed or similar).  This would make it clear what the trade off is. 

Literature review is OK, but needs a much clearer review of existing papers that compare remotely sensed gully measures with field measures.  Why do we need a new study?  You need to spell out accuracy of other methods so that you can compare them with this new one. 

Objectives of the paper are not at all clear.  They need to be spelled out more clearly.

Methods are confusing between field and desktop measurements.  Divide into these two areas. 
